# Status of Schistosomiasis Elimination in the Caribbean Region

**DOI:** 10.3390/tropicalmed4010024

**Published:** 2019-01-31

**Authors:** Reynold Hewitt, Arve Lee Willingham

**Affiliations:** One Health Center for Zoonoses and Tropical Veterinary Medicine, Ross University School of Veterinary Medicine, Basseterre, Saint Kitts and Nevis; reynoldhewitt@students.rossu.edu

**Keywords:** *Schistosomiasis mansoni*, Caribbean, elimination, snail control, *Biomphalaria glabrata*

## Abstract

Schistosomiasis elimination status in the Caribbean is reviewed with information on historical disease background, attempts to control it and current situation for each locality in the region where transmission has been eliminated (Sint Maarten, Saint Kitts, Vieques), eliminated but not yet verified (Puerto Rico, Dominican Republic, Antigua, Montserrat, Guadeloupe, Martinique) and still ongoing (Saint Lucia, Suriname). Integrated control initiatives based on selective and mass treatment and snail control using environmental, chemical and biological methods along with public service improvements (housing, safe water, sanitation) and changes in demography (urbanization) and economy (change from sugarcane and banana production to tourism) have resulted in reduction in the burden of schistosomiasis over the past century. Introduction of *Biomphalaria*-competitor snails into the region as a cost-effective, low maintenance control method appears to have had the most sustainable impact on transmission reduction. A regional inventory of *B. glabrata*, other *Biomphalaria* species and *Biomphalaria*-competitor snails as well as investigation of possible animal reservoir hosts in persisting endemic areas would be helpful for control. Elimination of schistosomiasis appears achievable in the Caribbean. However, a regional surveillance and monitoring program is needed to verify elimination in the various localities and identify and monitor areas still endemic or at risk.

## 1. Introduction

*Schistosoma mansoni* is the only human-infecting schistosome species that occurs in the Caribbean countries and territories (For the purposes of this article the Caribbean region is considered as all the islands in the Caribbean Sea and those adjoining mainland areas which are politically, historically and culturally associated with the islands (i.e., Belize, Guyana, Suriname, French Guiana) and is thus composed of sovereign states, overseas departments, and dependencies.). The first indication that *S. mansoni* was present in the Caribbean occurred in 1902 when Sir Patrick Manson published a case report [1] regarding the discovery of lateral-spined schistosome ova in the feces of a 38-year-old Caucasian Englishman who had been living in the Caribbean for 15 years at the time of his examination in England. The man had suffered chronic lower back pain and severe headaches over a period of 5 years, which at times had totally incapacitated him, as well as a history of multiple bouts of malaria. He was found to be anemic with eosinophilia and positive for intestinal schistosomiasis. The patient had never been to Africa, only having been in the British and Caribbean Isles during his lifetime. From 1887–1902 the patient had lived in Antigua, Anguilla and St Kitts, paid multiple visits to Nevis and Montserrat and also travelled to St Thomas and Barbados, thus it is difficult to pinpoint the exact island where he had been exposed to schistosomiasis. This first case report from the Caribbean was followed in 1904 by diagnosis of intestinal schistosomiasis in two young boys from the Mayagüez region of Puerto Rico who were found by Doctor I G Martínez to have lateral-spined schistosome ova in their feces [2].

Since the beginning of the 20th century schistosomiasis mansoni has been found endemic in several countries and territories in the Caribbean region, its discontinuous geographic distribution determined to a great extent by the occurrence of its intermediate hosts, freshwater snails of the genus *Biomphalaria* (Preston, 1910) [3]. Phylogenetic studies, supported by the fossil record, suggest that *Biomphalaria* actually originated in the Americas and secondarily colonized Africa within the past 5 million years [4]. Conversely, based on molecular evidence schistosomiasis mansoni in the Americas is considered to have come most likely from West Africa via the slave trade between the 17th to 19th centuries enabling transmission to the *Biomphalaria* snails present and establishing the life cycle of the parasite in the Caribbean and South America [5]. The significance of schistosomiasis mansoni as a zoonosis and the possible role of wild animals maintaining transmission in the Caribbean remains to be clarified with monkeys and rodents found with patent infections in various locations in the region during the past century [6,7,8].

An overview of the localities with a history of being endemic for schistosomiasis mansoni is provided in Figure 1. Two countries in the Caribbean region, Saint Lucia and Suriname, are still considered endemic for schistosomiasis with possible residual transmission whereas six additional countries and territories of Antigua and Barbuda, Guadeloupe, Martinique, Montserrat, Puerto Rico and Dominican Republic have likely eliminated transmission, but their status needs to be verified by compiling an elimination dossier and/or conducting epidemiological surveys based on WHO recommendations [9]. There are a few islands where schistosomiasis was endemic that became free of the parasite through cessation of transmission. Schistosomiasis cases ceased being detected on the island of Saint Martin/Sint Maarten from 1929 onwards with surveys published in 1980 indicating the absence of snail habitats on the island [3,10]. St Kitts was highly endemic in the early 1900s with more than 25% of the population estimated to be infected in 1932 but by 1959 onwards schistosomiasis was no longer considered a public health problem [11,12]. Schistosomiasis mansoni was eliminated from the small island of Vieques near Puerto Rico in 1962 [13]. Caribbean localities without a history of schistosomiasis include Anguilla, St Barthélemy, Saba, Sint Eustatius, Nevis, Dominica, St Vincent and the Grenadines, Grenada, Barbados, Tobago and Trinidad, Aruba, Bonaire, Curaçao and the British and U.S. Virgin Islands in the Lesser Antilles as well as the Greater Antillean countries/territories of Jamaica, Cuba, Cayman Islands and Haiti, nor in Belize, Guyana or French Guiana which are also included in the Caribbean region [3,14]. 

Transmission of schistosomiasis in the Caribbean was enhanced through agricultural production, whereby irrigation and drainage systems developed for production of sugarcane, bananas and rice resulted in expansion of habitats for *Biomphalaria* snails and increased exposure of people to water harboring *S. mansoni* cercariae while movement of aquatic plants for ornamental or piscicultural use enabled spread of potential intermediate-host snails between localities in the region [3,14]. Various interventions including hydrological changes, improvement in water and sanitation, targeted chemotherapy, and intermediate-host snail control have been undertaken in the region with varying degrees of success in lowering the burden of schistosomiasis and blocking its transmission. The St Lucia Project undertaken between 1965 and 1981 with support from the Rockefeller Foundation and the local government actually enabled a large-scale, concurrent comparison study of drug-treatment versus intermediate-host snail control versus improvement of water supplies [15,16].

Our aim is to review the schistosomiasis situation in each of the localities of the Caribbean where schistosomiasis mansoni has been and may still be present to better understand the lessons learned and identify continued research and control needs that would help enable elimination of the disease from the entire region. 

## 2. Status of Schistosomiasis in the Caribbean

### 2.1. Localities Where Transmission of Schistosomiasis Mansoni Eliminated

#### 2.1.1. Saint Martin

In the 1920s over 20 cases of schistosomiasis and multiple snail habitats were detected near a sugar mill in Colombier, considered the wettest valley in the French half of the French/Dutch island of Saint Martin/Sint Maarten [10,11]. No further autochthonous cases have been reported on the island since 1929 [14]. This was attributed to the disappearance of snail habitats as a result of the climate becoming drier and because of extensive deforestation associated with major construction projects resulting in hydrological changes such that surface water from heavy rain drains away quickly [3]. 

#### 2.1.2. Saint Christopher (Saint Kitts)

The island of St. Kitts was highly endemic for schistosomiasis in the first half of the 20th century with multiple cases reported in 1918 [6] and a 1932 survey based on single saline fecal smears indicating nearly 25% of the island’s population infected [11]. Schistosomiasis cases were primarily found in coastal villages bordering the two permanent streams on the island, Wingfield River entering the Caribbean Sea at the village of Old Road and Cayon River entering the Atlantic Ocean at Cayon village, as well as a semi-permanent stream appearing during heavy rainfall which entered the Caribbean Sea at the village of West Farm but during other times only provided piped water to the plantation of West Farm itself. The drinking water at Cayon was piped from a mountain spring free from snails; but the drinking water to Old Road and West Farm, together with the washing/bathing water in all the endemic areas, was not free of snails [17]. Importantly, St Kitts is inhabited by West African Green (Vervet) monkeys (*Chlorocebus aethiops*) and five of seven of these monkeys collected near these streams in 1928 were found infected with schistosomiasis [6].

In the 1940s, a scheme was introduced to secure domestic water supplies by diverting the natural mountain streams such that aquatic snail habitats were destroyed in the low-lying areas as water no longer flowed constantly to those areas [12]. These hydrological changes affected the *B. glabrata* habitats resulting in interruption of schistosomiasis transmission as evidenced by a steep decrease in incidence such that autochthonous clinical cases in St. Kitts have not been detected since 1955 [3]. In 1959 uninfected *B. glabrata* were found to persist in West Farm Gut [3] instigating a randomized survey of 188 randomly selected school children and villagers from the West Farm area with no cases detected [12]. Attempts were made to eliminate *B. glabrata* from the island by applying a molluscicide (Bayluscide) to relevant water bodies in 1965 and 1976 and introducing *Marisa cornuarietis*, the Colombian ramshorn apple snail, as a biological control agent in West Farm Gut [3]. A *B. glabrata* colony was later detected in the isolated Fountains River in the rain forest at an altitude of 450 meters [3]. There is currently no active schistosomiasis surveillance program on the island. Fecal samples collected in 2015 from 94 wild-caught monkeys and subjected to copro-PCR testing at the WHO Collaborating Centre for the Identification and Characterisation of Schistosomes and Snails at the Natural History Museum in London indicated no positives (Jennifer Ketzis, Ross University School of Veterinary Medicine, personal communication).

#### 2.1.3. Vieques

Vieques Island lies off the east coast of Puerto Rico, and as part of the Commonwealth of Puerto Rico is a U.S. territory. In 1954 the prevalence of schistosomiasis in 6-year-old children on the island was 6.7% when a concerted intense control program involving control of snails with molluscicide (sodium pentachlorophenate) and chemotherapy of infected persons (sodium antimony tartrate) was implemented by the Puerto Rico Health Department with technical assistance from the U.S. Public Health Service [13]. By 1958 transmission had stopped among children in schools at high risk, and by 1959 the prevalence was down to zero and has remained there [14]. No autochthonous cases of schistosomiasis have been detected on the island since 1962. Total elimination of *B. glabrata* snails was not achievable; however, the great reduction in their number enabled transmission to be interrupted. The island remains free of schistosomiasis although *B. glabrata* persists.

### 2.2. Localities Where Schistosomiasis Transmission Considered Eliminated but Not yet Verified

#### 2.2.1. Puerto Rico

Schistosomiasis was first detected on the island of Puerto Rico in1904 when young boys from Mayagüez region were found to have lateral-spined schistosome eggs in their feces and 21 cases of intestinal schistosomiasis were detected by stool survey of 4482 anemic patients from Utuado [2,18]. In 1913, the Institute of Tropical Medicine of Puerto Rico recorded 320 cases of bilharziasis among 10,149 patients [18]. Surveys in the 1930s indicated schistosomiasis cases across the country, with the eastern part of the country having the highest disease prevalence, in particular the heavily irrigated sugarcane-producing region of Guayama-Arroyo-Patillas [19,20,21]. An island-wide survey in 1943 estimated prevalence at 13.5% while a coprological survey conducted in 1950 on 11,690 schoolchildren between 5 and 18 years of age in 17 municipalities indicated a prevalence of 10% [18]. Schistosomiasis was prevalent along the island’s coasts, along lowlands and valleys of the interior where sugarcane is grown and in one mountainous region (Utuado) [18].

In 1943, a national program was instituted to improve availability of water and sanitation through construction of water supply and sewage systems including the channeling of streams through enclosed cement viaducts that likely resulted in reduction of schistosomiasis transmission [20]. A national schistosomiasis control program under the Puerto Rico Health Department was instituted from 1953 to 1980 that emphasized biological, environmental and chemical control of snails as well as health education, improvement of public water supplies, free latrine distribution and limited chemotherapy [20,22,23]. Chemotherapy was initially not widely used due to concerns about dangerous and deadly side effects of available drugs during that time [20]. The control program first used copper sulfate for mollusciciding but soon switched to sodium pentachlorophenate and eventually niclosamide in the 1960s and 1970s [20]. 

Between 1954 and 1958 the *Biomphalaria*-competitor snails *M. cornuarietis* and *Tarebia granifera* appeared in Puerto Rico and within a decade spread throughout the country effectively displacing the *B. glabrata* populations with profound impact on schistosomiasis transmission [24,25]. In 1956, *M. cornuarietis* was intently transferred to 111 irrigation ponds containing *B. glabrata* in the south part of the island and by 1965 was found to have completely displaced the *B. glabrata* populations in 89 of the 97 ponds still in operation [26,27,28]. In 1997, a national survey of multiple water bodies known to be habitats for *B. glabrata* revealed the presence of *T. granifera* and *M. cornuarietis* but *B. glabrata* had disappeared [24]. Because of the success of competitor snail introductions, biological control was found to be far more cost-effective and sustainable than mollusciciding and thus given priority [29].

An intense control program was undertaken from 1954–1960 in the highly endemic community of Patillas which included health education in primary schools and rural communities, limited chemotherapy, monitoring surveys based on annual fecal testing (formalin-ether concentration) of 1st graders and chemical, environmental and biological control of snails using sodium penta chlorophenate, drainage of habitats and introduction of the predatory *Biomphalaria*-competitor ampullarid snail *M. cornuarietis*, respectively [30]. The program was a joint effort of the Puerto Rico Department of Health and San Juan Laboratories of the Public Health Service, involving biologists, engineers and physicians. At an average annual cost of $8600, the program was effective such that by 1962 prevalence of schistosomiasis among 7-year-old children dropped from 21.5% in 1952 to 0% and the *B. glabrata* snail population no longer existed [30]. 

The national schistosomiasis control program with its main focus on biological snail control without widespread human treatment showed slow progress initially but had very positive results in the long term. Nationwide prevalence decreased from 15.6% in 1963 to 9% in 1968 [31] to 3.4% in 1975 [32]. By 1976, only 5 of the island’s 30 main freshwater reservoirs still harbored *B. glabrata* [33]. Recognizing this progress and facing new challenges the Puerto Rico Department of Health eliminated the schistosomiasis control program in 1980 and transferred its resources to combating dengue virus. There was little in the way of active control efforts in the 1990s however nationwide disease prevalence further decreased to 2% in 1989 [34] and a limited survey in 1997 [19] conducted in previously endemic areas revealed only three positive cases all in older individuals. During this time major demographic changes happened on the island as a result of rapid urbanization with the rural population migrating to cities such that the rural population decreased from 59.5% to 6.4% of the total population between 1950 and 2018 [35,36] greatly reducing the number of persons exposed in rural transmission sites while the island’s sanitation and health systems were strengthened. Authorities now consider schistosomiasis transmission interrupted as there are no longer clinical cases and no cases have been detected from coprological or serological surveys in recent years however compilation of an elimination dossier and follow-up studies, per WHO recommendations, are still needed to verify the status [9,18]. 

#### 2.2.2. Dominican Republic

The first undisputed autochthonous schistosomiasis case in Dominican Republic was reported in 1947 [37]. Historically, active transmission occurs in the eastern part of the country primarily in the provinces of El Siebo, La Altagracia and Hato Mayor, the latter considered the area of highest endemicity [38]. Cases have also been reported further north in Sabana de la Mar and Miches [37]. The area is known for sugarcane and rice production and this has been linked to the high transmission as have the frequent domestic and recreational use of streams and consequent fecal contamination [39]. Infected *B. glabrata* snails were first detected in this area in 1951 [18]. Surveys in the 1960s indicated the geographic range of *B. glabrata* snails to be more extensive than that of schistosomiasis transmission with *B. glabrata* found present in over 1/6th of the country’s land area including the capital Santo Domingo [38]. 

Schistosomiasis control efforts commenced in 1952 initially focused on mollusciciding snail habitats in high transmission areas with sodium pentachlorophenate [40]. In 1970 a Committee to Combat and Control Bilharzia, located in Hato Mayor, was created within the Ministry of Public Health and Social Welfare and a Center for the Eradication of Bilharzia established to serve as a base for implementing and coordinating control efforts [39]. The control program’s activities included community education, schistosomiasis surveys involving coprologic detection, environmental control (e.g., draining ponds, filling-in swamps), and snail control based on molluscicide application (Frescon, Bayluscide) as well as biological control using both ampullarid and thiarid competitive snails [39]. In the 1960s and 1970s *M. cornuarietis*, *T. granifera* and *Melinoides tuberculata* were both actively and passively introduced into waterways in the high transmission region all becoming prolific in streams, irrigation canals and drains in rice plantations such that in several locations they completely displaced the *B. glabrata* snails [39]. In 1980 surveillance efforts were taken over by the Bilharzia Institute established through a national resolution under the Autonomous University of Santo Domingo as a public health program to undertake numerous epidemiological and malacological studies until dissolved in 1996 [18].

In 1968, there were an estimated 1000 infected individuals and 6000 at risk nationally [41]. By 1986, these numbers had increased dramatically to 208,000 people infected and an estimated 4.16 million people at risk, with an estimated 5% prevalence [34]. In 2008 estimates indicated 258,000 people to be infected [42] and in 2010 prevalence estimated at 3.0% [43]. Evidence suggests prevalence has decreased significantly in the past few years as a result of urbanization, economic development, improvements in sanitation, and expanding presence of competing snails [39]. In 2013 a serological survey (ELISA and immunoblot) was conducted in the provinces with a history of schistosomiasis transmission with none of 612 samples collected testing positive [18]. The current objective is to update knowledge of the presence and distribution of *Biomphalaria* spp. snails while compiling all of the necessary evidence to verify that transmission has indeed been interrupted through post-elimination surveillance [44].

#### 2.2.3. Antigua

Schistosomiasis cases were first detected on the island of Antigua in 1923 when 18% of persons in St John Parish included in a Rockefeller Foundation hookworm survey were found passing eggs of *S. mansoni* [3]. *Biomphalaria glabrata* snails were found in St John Parish in Bendel’s stream in 1928 and later in nearby Body Pond, with these waterbodies continuing to serve as important transmission sites for several decades due to domestic and recreational use [3]. A 1932 report recorded 60% prevalence in a village bordering these water bodies [11]. Official surveillance and control programs were never established in Antigua as transmission was considered low such that malacological and schistosomiasis surveys have been infrequent. A 1977 survey showed *B. glabrata* to be widespread and abundant in pools, dams and canals throughout the island but human contact with them limited [3]. In 1980, the *Biomphalaria*-competitor snail, *T. granifera*, was observed on the island and later *M. tuberculata* also appeared [45]. 

In the 1980s hydrological changes as a result of several dam projects resulted in the disappearance of any permanent flowing bodies of water as the water was piped to households for domestic use which improved sanitation and also greatly reduced exposure of the population to transmission sites as bathing and washing were increasingly done at home [14]. These water impoundment projects resulting in creation of dams and reservoirs with concomitant improvements in sanitation and protection of water supplies had great impact on the incidence of schistosomiasis [3]. National surveys in 1989, 2000 and 2003, estimated schistosomiasis prevalence at 0.1–0.2% [34,43,46] resulting in the island no longer being considered endemic however compilation of an elimination dossier and snail and schistosomiasis surveys are still needed to verify this status.

#### 2.2.4. Montserrat

Montserrat was reportedly clear of schistosomiasis in the first half of the 20th century with no autochthonous cases reported before 1947 [41]. In 1974, public health officials reported low endemicity of schistosomiasis but provided no data. A 1977 survey indicated the presence of *B. glabrata* snails in the small, sparsely inhabited watersheds of Barzey’s Stream where local people washed clothes and bathed and Farms River a local swimming site [3]. In 1978, researchers surveyed schistosomiasis prevalence in the few individuals living close to these two localities finding 14% positive while their survey of 251 school-age children elsewhere on the island did not indicate any positives [47]. In 1989, the WHO estimated that only 145 people out of the island’s 11,200 were at risk of contracting schistosomiasis (1.3% at risk) and later removed Montserrat from the list of schistosomiasis endemic countries in 1993 [34,48]. A major volcanic eruption in 1995 buried the two endemic areas of the island thus interrupting schistosomiasis transmission completely. However, due to a lack of data WHO has reinstated the island’s place on the list of countries for which the epidemiological status of schistosomiasis remains ‘uncertain’ [48]. The Thiarid snail *M. tuberculata*, used as a biological competitor to control *B. glabrata* on other islands, was found on Montserrat in 2001 [49]. 

#### 2.2.5. Guadeloupe

Guadeloupe consists of two islands separated by a narrow strait with the western island, Basse-Terre, being volcanic, mountainous, forested, and containing numerous permanent rivers, streams and canals while the eastern island, Grande-Terre, is low-lying, flat, calcareous, well-drained, and has only temporary streams and natural and artificial ponds and marshes. Cases of schistosomiasis have been reported from the entire island of Grande-Terre, with mangrove swamps important transmission sites. However, more cases were reported from the coastal belt of mountainous Basse-Terre with streams and irrigation canals major sites of transmission [3]. Prevalence of *S. mansoni* peaked in the 1960s and 1970s such that in 1978 human prevalence on the islands was estimated to be 25% [25]. 

An interesting epidemiological aspect of schistosomiasis on Guadeloupe is the finding of rats serving as definitive hosts of *S. mansoni* with both the black rat, *Rattus rattus*, and brown rat, *Rattus norvegicus,* found naturally infected [8]. At two localities in particular, the mangrove swamps of Grande-Terre and the Grand Etang Lake in the rain forest of south Basse-Terre, rats were found to maintain the life cycle of *S. mansoni* alone in the absence of human infection with both high prevalences and high intensities of infection with some rats harboring more than 500 worms [50]. The *S. mansoni* cercariae in these areas were shown to have a late shedding pattern with an emergence peak adapted to the nocturnal activities of the rodents [51]. Studies indicate that that murine reservoir hosts may not be an important factor in areas where human transmission occurs [52]. 

In 1978, Guadeloupe established an integrated control program based on treatment of infected persons, environmental management and snail control mainly involving introduction of *Biomphalaria*-competitive snails (*Pomacea glauca* in 1976, *M. cornuarietis* in 1987, *M. tuberculata* in 1985) [25]. Prevalence was reduced to 15% in 1985 [34] and most of the transmission sites were eliminated in the1990s due to the extensive displacement of *B. glabrata* by *M. tuberculata* throughout most of the islands such that prevalence was down to 1% in 2003 [43,53]. *Melanoides tubercalata* was unable to effectively displace but ended up co-existing with *B. glabrata* in the mangrove swamps [54] where murine schistosomiasis was still present in *R. rattus* in 2005 [53]. Authorities consider schistosomiasis transmission on Guadeloupe effectively interrupted however, its status awaits evaluation and verification by the World Health Organization. 

#### 2.2.6. Martinique

*Biomphalaria glabrata* snails were included in an inventory of malacological fauna of Martinique already in 1873 [55] while the first cases of schistosomiasis were detected in 1906 when native persons being treated for dysentery were found passing lateral-spined schistosome eggs in their feces [56]. The prevalence of schistosomiasis was estimated at 6.4% in 1951, 8.4% in 1961 and 12% in 1977 [57]. Important sites of transmission were beds of watercress where people became infected while collecting the vegetation for consumption [58]. These surveys led to recognition of schistosomaisis as a public health problem resulting in establishment in 1978 of a national Department to Combat Intestinal Parasitoses to implement integrated control efforts including sanitation improvements both individually and collectively, health promotion and education, detection and treatment of patients (niridazole from 1978, oxamniquine from 1981, and praziquantel from 1995) and snail control [18]. However, unintended changes in the malacological fauna of the island had profound and sustained impact on schistosomiasis transmission such that the control program eventually became focused on biological control as its long-term strategy.

A 1953 malacological survey of the island reported *B. glabrata* was widespread [59]. During the late 1960s and early 1970s, *Biomphalaria kuhniana* (initially reported as *B. straminea*) entered the island and gradually displaced *B. glabrata* such that *B. kuhniana* rapidly became distributed across the whole hydrogeographic system of the island [60]. Although *B. kuhniana* was compatible as an intermediate-host of *S. mansoni* it was less susceptible to infection than *B. glabrata* which resulted in reducing transmission [61]. In 1979, the *Biomphalaria*-competitor thiarid snail, *M. tuberculata*, which is not susceptible to *S. mansoni*, entered Martinique and rapidly colonized the entire island displacing populations of both *B. glabrata* and *B. kuhniana* as a result reducing the prevalence of schistosomiasis [61].

The success of the accidental introduction of *M. tuberculata* in interrupting schistosomiasis transmission led the Ministry of Health to initiate a biological control program against schistosomiasis, using *M. tuberculata* in 1983 [61]. By 1989, after several intentional introductions of *M. tuberculata* into watercress beds and other transmission sites, the prevalence of *S. mansoni* dropped to 1.33% and by 1990 malacological surveys proved the near total disappearance of both *B. glabrata* and *B. kuhniana* [62]. In the 1990s another thiarid *Biomphalaria*-competitor snail. *T. granifera*, entered Guadeloupe and rapidly spread throughout the island partially displacing *M. tuberculata* [63]. Together the thiarid snails have colonized the whole hydrographic system of the island and maintain dense populations preventing an eventual recolonization by the *Biomphalaria* spp. snails and thus are maintaining sustainable control of schistosomiasis [25]. Disappearance of the intermediate hosts of *S. mansoni* and the absence of any new cases of infection in children under 10 years of age since 1984 support the elimination of schistisomiasis from Martinique. However, this remains to be verified through malacological and human prevalence studies [18].

### 2.3. Localities Where Schistosomiasis Transmission Still Considered Ongoing

#### 2.3.1. Saint Lucia

Schistosomiasis in St Lucia was first reported in the 1924–1925 period [64]. Although *B. glabrata* were found widespread on the island, schistosomiasis initially remained focal, contained primarily in the southwestern Soufriere Valley until the 1950s when there was a shift in agricultural production from sugarcane to bananas and the *B. glabrata* expanded into the banana root drainage channels in the valleys and the extensive network of associated natural river systems which people used intensively for domestic and recreational use [14,65]. These environmental conditions and socioeconomic determinants including limited water and sanitation services leading to unsafe drinking water, outdoor defecation, and water contact for bathing and washing clothes contributed to transmission [66]. By 1961, the estimated national prevalence based on coprological testing was 17%, and evidence indicated that *S. mansoni* had spread to most parts of the island [15]. By the 1970s schistosomiasis prevalence in many of the endemic communities reached 70% [3].

Between the years 1965 and 1981, the St. Lucia Research and Control Project, a joint effort of the St. Lucia Ministry of Health and the Rockefeller Foundation with financial assistance from the Medical Research Council of the United Kingdom, utilized the natural laboratory setting of an endemic island to conduct research on all facets of schistosomiasis including the biology of *B. glabrata*, treatment of humans and the best methods for controlling transmission [15]. The main focus of the project was on the basic schistosomiasis control methods, namely health education, provision of safe water, chemotherapy and snail control, investigating through a comparative design the impact of these methods separately and in combination in ecologically isolated watersheds throughout the island, referred to as the “experimental valley” approach [4,15]. For chemotherapy those found infected were initially treated with hycanthone in the early 1970s and then with oxamniquine after 1975 [15]. Mollusciciding, using an emulsifiable concentrate of 25% niclosamide, was first applied area-wide and monitored by snail surveillance, then followed up with focal treatment if *B. glabrata* was still found present [65,67,68]. In 1978, the Project intentionally introduced the *Biomphalaria*-competitor snail, *M. tuberculata*, which quickly colonized water bodies on the island resulting in various levels of displacement of *B. glabrata* populations [15]. Surveillance of transmission was achieved using a locally developed sentinel-snail technique [69] and a coprological sedimentation and concentration technique for detecting human infection [70]. Incidence of schistosomiasis was reduced in communities where interventions were implemented compared to control communities [70,71,72,73,74]. When the Project ended in 1981, the risk of disease was considered minimal and prevalence at known endemic areas was less than 2 % [14].

After the joint Ministry of Health–Rockefeller Foundation St. Lucia Research and Control Project ended in 1981 parasite surveillance and control activities were very limited but indicated that schistosomiasis was not totally eliminated from the island [66]. Between 1981–1986 *M. tuberculata* was introduced to the whole hydrographic system of St Lucia. However, in 1993, *B. glabrata* was still found co-existing with *M. tuberculata* in 17 sites and abundantly present in two other sites where *M. tuberculata* was absent, and another *Biomphalaria* species, *B. straminea*, detected on the island [75]. Between 1995 and 2007, 106 schistosomiasis cases were reported mostly through passive surveillance of patients at prenatal health care centers and among food handlers who were routinely tested [66]. A coprological survey of 10,508 people undertaken between 2002–2005 indicated 0.3% prevalence overall with no children under the age of 9 found infected though a prevalence of 0.5% was indicated in children 10–19 [76]. The last data on schistosomiasis in St Lucia came from a limited school-based coprological survey conducted in 2006 of 550 children in three southern, rural villages that detected schistosomiasis in four (0.6%) children between the ages of 5–14 [77] and detection of a case in Babonneau in 2007 [18]. Efforts are currently underway by the St. Lucia Ministry of Health and PAHO to undertake national surveys focusing on children and snails to determine the island’s current schistosomiasis status. As a pilot research site, St. Lucia is still considered a model for an integrated approach to schistosomiasis control; however, it also serves as an important example of the long-term needs for surveillance and monitoring and continued application of control measures to achieve complete elimination [43].

#### 2.3.2. Suriname

*Biomphalaria glabrata* was first reported in Suriname in 1859 while the first schistosomiasis case was detected in 1911 [66,78]. Schistosomiasis is endemic only in the coastal region of the country where the majority of the country’s population resides, stretching from the marsh areas north of Wageningen in Nickerie district in the west through the central cultivated areas surrounding Paramaribo to the delta area of Commewijne district in the east [66]. *Biomphalaria glabrata* are found in rice fields, swamps, drainage ditches and canals, their presence and distribution associated with the calcareous sands and alkaline waters of the shell-ridges [79]. Conversion of swamps to rice paddies in the region increased transmission as a major difficulty in control is the lack of proper disposal of feces of people working in rice fields and the irrigation schemes for rice cultivation expanded snail habitats [80,81]. 

The burden of schistosomiasis in Suriname has been explored through multiple studies. A house-to-house survey in 1956 based on fecal smears of over 10,000 people in multiple coastal plain localities indicated 12.7% excreting *S. mansoni* eggs [79]. A similar survey conducted from 1961–1964 focused on Saramacca district showed 23.1% positive [82] but this number went up to 45% by 1974 [66]. Intermittent national surveys conducted in the period 1986–2008 continually indicated about 3400–4000 people infected nationally [34,46]. A survey of schoolchildren in coastal provinces from 1997 to 2001 found between 0.3% and 4.7% prevalence [83]. Surveys conducted by the Ministry of Health in Commewijne and Saramacca in the period 1997–1998 indicated prevalences of 3.1% and 4.7%, respectively [81]. Countrywide prevalence remained between 0.9% to 1.0% in the period 2003–2010 [43]. High prevalences have been found associated with occupational activities involving water contact such as fishing and agriculture as well as among certain ethnic groups living in rural areas of all the districts of the coastal plain [80]. Studies conducted in the 1990s showed that schistosomiasis was a major cause of morbidity and mortality, with hematemesis, and pulmonary hypertension being the major consequences of infection [81]. 

Since the 1970s Suriname has implemented schistosomiasis control programs resulting in reduced prevalence of the disease. Initially hycanthone was used effectively to treat schistosomiasis however the drug had significant side effects, afterwards oxamniquine was used until 1983 when it was replaced with praziquantel [84]. The very high schistosomiasis prevalence of 74% detected in Saramacca in 1974 instigated a long-term control project in that district funded by the governments of both Suriname and the Netherlands aimed at reducing prevalence to 5% [85]. Under the project selective chemotherapy was used based on coprological case detection and implementation of other control activities including mollusciciding, mechanizing rice culture, installing pit latrines, draining standing water and swamps, and providing health education regarding schistosomiasis [86]. The first phase of the project was conducted 1974–1983 and then transferred to local authorities and integrated with existing public health programs [85]. National intervention efforts continue including intense information campaigns, detecting and treating infected people through stool sampling and provision of praziquantel, improved sanitation/waste disposal and avoidance of contact with contaminated water. A 2011 survey of six coastal districts and one inland district showed very low prevalence rates, well below the 20% deemed necessary to institute mass drug administration [87].

## 3. Research and Control Needs

### 3.1. Updating Epidemiological Surveillance

The 2009 PAHO Resolution CD49.R19 on elimination of neglected diseases and other poverty-related infections noted that in the Caribbean schistosomiasis is still present in Saint Lucia and Suriname and stated the need for studies to confirm its elimination from previously endemic localities, the goal being to reduce prevalence and parasite load in high transmission areas to less than 10% prevalence as measured by quantitative egg counts [91]. In 2012 the World Health Assembly adopted resolution WHA65.21 on elimination of schistosomiasis, calling for increased investment in schistosomiasis control and support for countries to initiate elimination programs [92]. As presented in this review, limited evidence suggests low or no transmission in many localities previously endemic for schistosomiasis including Dominican Republic, Puerto Rico, Antigua, Montserrat, Guadeloupe and Martinique. The epidemiological status of these historically schistosomiasis endemic localities in the Caribbean, in particular the prevalence and intensity of *S. mansoni* infections in children as an indirect indicator, needs to be updated based on WHO guidelines to verify elimination of transmission or, if schistosomiasis is still present, determine the public health interventions needed for control which can lead to elimination [9]. 

### 3.2. Regional Snail Inventory

A major component of the effort to combat schistosomiasis in the Caribbean region has been the effective and sustainable control of snails through environmental, chemical and, most importantly, biological interventions resulting in profound reduction or absence of *B. glabrata* populations and consequently *S. mansoni* in many sites. Biological control using the phenomenon of competition by non-medically important snails was found to be a cost-effective, low maintenance, popular way of controlling the disease. Competitive action of the ampullarid snail *M. cornuarietis* through food and predation of eggs and young snails [93] or *B. glabrata*’s avoidance of the prolific and rapidly colonizing parthenogenic thiarid snails *T. granifera* and especially *M. tuberculata* [94] all resulted in varying levels of displacement of *B. glabrata* populations. Biological control does have its limits as shown in St Lucia where *B. glabrata* and *M. tuberculata* were found to co-exist in many habitats [75], and in Guadeloupe the Thiarid snails were unable to displace *B. glabrata* in certain areas where rats served as *S. mansoni* intermediate hosts [54]. In addition, other species of *Biomphalaria* including *B. straminea*, *B. havenensis* and *B. kuhniana* have been found present in the region with varying susceptibility to *S. mansoni* [60,75,78]. With the current aim to eliminate schistosomiasis from the region, it would be worthwhile to now undertake an inventory of *Biomphalaria* spp. as well as *Biomphalaria*-competitor snail species in the Caribbean, to help assess the long-term impact of biological control, determine areas still at risk for schistosomiasis transmission, investigate the potential of other *Biomphalaria* spp. as transmitters of *S. mansoni* and contribute to the verification of schistosomiasis elimination.

### 3.3. Animal Reservoir Hosts

With the aim to eliminate schistosomiasis from the Americas, the presence and importance of animal reservoir hosts of *S. mansoni* becomes a significant issue for the eventual success of elimination efforts as they could pose a permanent threat of recrudescence of human infection [14]. African Green Monkeys, *Chlorocebus aethiops,* in St Kitts were found infected with *S. mansoni* [6] though apparently dependent on the persistence of human infection as after abatement of schistosomaisis transmission to humans, primate infections were no longer observed [14]. The occurrence of schistosome infections in multiple mammal species in Suriname was considered to have similar incidental status [7,95]. Under the Schistosomiasis Research and Control Project in St Lucia numerous animal species including mammals and birds were tested for infection with none found positive though in many cases the sample sizes were limited (15). The importance of rats as reservoir hosts maintaining transmission of schistosomiasis in Guadeloupe specifically in standing-water habitats where rat-maintained *S. mansoni* populations could persist independent of human intervention has been identified [96,97]. The real significance of murine schistosomiasis in Guadeloupe is not in its current impact on human schistosomiasis transmission but its potential for confounding control and eventual elimination efforts [14]. In areas of the Caribbean region with continued schistosomiasis transmission, e.g., Suriname and St Lucia, the presence and importance of animal reservoir hosts should be investigated to ensure that persistence of the disease is not due to that reason. 

## 4. Conclusions

During the 20th Century schistosomiasis mansoni emerged as an important cause of morbidity and mortality throughout the Caribbean region however, at the beginning of the 21st Century the burden of disease appears dramatically reduced and confined to only a few foci such that its importance as a public health problem has greatly diminished. Integrated control programs based on selective and mass treatment campaigns and snail control through environmental, chemical and biological interventions enabled reductions in prevalence over the decades as evidenced in Puerto Rico and St Lucia. Additionally, the socioeconomic and ecological determinants that enabled schistosomiasis transmission have been impacted dramatically during this time due to demographic changes and improvement in socioeconomic standards resulting in urbanization, improvements in housing, provision of safe water and sanitation and re-focusing of economies from being based on agricultural production (sugarcane, bananas) to tourism that have resulted in reducing both people’s exposure to *S. mansoni* as well as outdoor defecation subsequently limiting transmission. One of the most important factors responsible for long-term, sustainable impact on transmission has been the appearance of *Biomphalaria*-competitor snails in the region, both intentional and unintentional, particularly the rapidly colonizing oriental thiarid snail *M. tuberculata*. Based on these factors the elimination of schistosomiasis from the Caribbean region should indeed be achievable (see Table 1). A regional surveillance program is now needed, led by PAHO, CARPHA, CARICOM and local governments to identify in a standardized way remaining endemic foci as well as areas still at risk using the most sensitive methods available.

## Figures and Tables

**Figure 1 tropicalmed-04-00024-f001:**
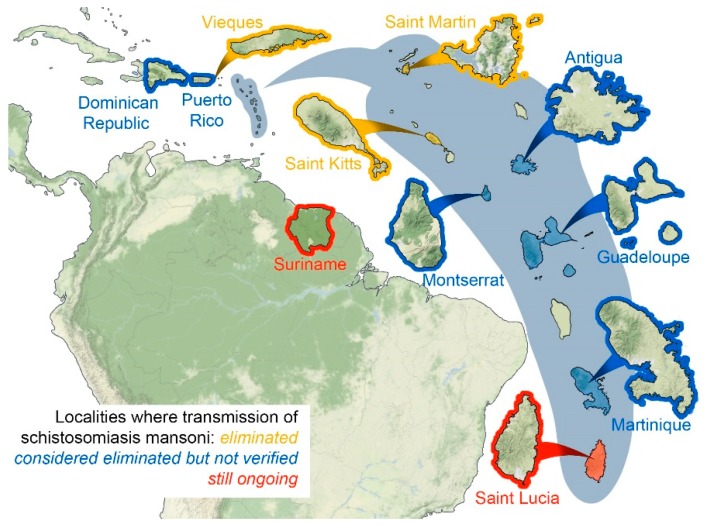
Map of the Caribbean region indicating localities with a history of schistosomiasis mansoni endemicity and their current transmission status.

**Table 1 tropicalmed-04-00024-t001:** Transmission status of Caribbean localities with a history of schistosomiasis endemicity noting interventions implemented and natural changes affecting transmission as well as presence of animal reservoirs.

Transmission Status.	Locality	Current Population ^1^	Interventions/Natural Changes Impacting Transmission	Animal Reservoirs
Eliminated	Saint Martin	32,284 (2018)	Deforestation Hydrological changes Climate change	
	Saint Kitts	34,918 (2011)	Hydrological Changes Water and sanitation improvement Mollusciciding Competitor snails (A) ^3^	Monkeys ^4^
	Vieques	8669 (2017)	Mollusciciding Chemotherapy	
Considered eliminated but not yet verified	Puerto Rico	3,337,177 (2017) ^2^	Water and sanitation improvement Chemotherapy (selective) Mollusciciding Competitor snails (A,B) ^3^ Health education	
	Dominican Republic	10,649,000 (2016)	Mollusciciding Health education Competitor snails (A,B,C) ^3^ Environmental management	
	Antigua	90,755 (2015)	Hydrological changes Water and sanitation improvements Competitor snails (B,C) ^3^	
	Montserrat	5241 (2015)	Volcanic eruption Competitor snails (C) ^3^	
	Guadeloupe	402,119 (2013)	Chemotherapy (selective) Environmental management Competitor snails (A,C,D) ^3^	Rats ^5^
	Martinique	385,551 (2013)	Water and sanitation improvements Health education Chemotherapy (selective) Competitor snails (B,C) ^3^	
Ongoing	Saint Lucia	172,255 (2014)	Health education Water and sanitation improvements Chemotherapy (selective) Mollusciciding Competitor snails (C) ^3^	
	Suriname	541,638 (2012)	Chemotherapy (selective) Mollusciciding Environmental management Water and sanitation improvements Agricultural practices Health education	

^1^ For Puerto Rico and Vieques: [88]; for Saint Martin: [89] ; for all others: [90]; ^2^ Puerto Rico population number includes population of Vieques; ^3^ Competitor snails: A = *Marisa cornuarietis*; B = *Tarebia granifera*; C = *Melanoides tuberculata*; D = *Pomacea glauca*; ^4^ African Green (Vervet) Monkeys (*Chlorocebus aethiops*); ^5^ Black and Brown Rats (*Rattus rattus* and *R. norvegicus*)

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
