# Peer review of "Status of Schistosomiasis Elimination in the Caribbean Region"

_tropicalmed, 2019, doi:10.3390/tropicalmed4010024_

Round 1
Reviewer 1 Report
The manuscript describes the present situation of schistosomiasis in the Caribbean Islands, pointing out the control achievements against the disease in the Region.
This is a very interesting MS, from historical background to research perspectives, including the potential role of rodents on S. mansoni transmission, which is quite relevant for all those involved in schistosomiasis control in endemic areas. Most importantly, it demonstrates the feasibility of schistosomiasis elimination or effective reduction when social, educational, therapeutic end environmental interventions are implemented together. Of particular note, the success of competitor snails, a biological approach with long-lasting positive effects and safe to hydrological environments.
The MS is clear and well written. I recommend the publication of the MS after minor corrections:
1. Line 128: “erase prevalence” (repeted)
2. Line 359: “… was first applied area-wide..” – please, correct this sentence.
3. References should be uniformed as some authors or publications are cited differently or incorrectly written.
Some examples: Refs. ; 16; 35; 45; 50; 57; 67; 68; 89.
Author Response
We appreciate the Reviewer’s comments regarding the MS!
1. Line 128: “erase prevalence” (repeted)
We appreciate the Reviewer noting this error and have revised the text accordingly:
In 1954 the prevalence of schistosomiasis in 6 year-old children on the island was 6.7% when a concerted intense control program involving control of snails with molluscicide (sodium pentachlorophenate) and chemotherapy of infected persons (sodium antimony tartrate) was implemented by the Puerto Rico Health Department with technical assistance from the U.S. Public Health Service [13].
2. Line 359: “… was first applied area-wide..” – please, correct this sentence.
We thank the Reviewer for noting this error and have revised the text accordingly:
Mollusciciding, using an emulsifiable concentrate of 25% niclosamide, was first applied area-wide and monitored by snail surveillance, then followed up with focal treatment if B. glabrata was still found present [65,67-68] .
3. References should be uniformed as some authors or publications are cited differently or incorrectly written.
Some examples: Refs. ; 16; 35; 45; 50; 57; 67; 68; 89.
We appreciate the Reviewer informing us of these editorial needs. We were able to find the error with references 35,45, 57, 67 and 68 and made the necessary corrections. We could not find errors with Refs 16, 50 and 89 but are ready to make those corrections if informed what is needed.
Reviewer 2 Report
This manuscript presents a well-researched and written paper on the current status of the schistosomiasis prevalence and the status of elimination of the infection and disease in Caribbean region.
Author Response
Thank you
Reviewer 3 Report
The manuscript tried to review the schistosomiasis situation in each of the localities of the Caribbean, to better understand the lessons learned and identify continued research and control needs. This work has a significant contribution to the field . Specific comments are listed below.
The only figure named ‘status of schistosomiasis mansoni in Caribbean’ may be better presented as a map with more comprehensive information such as geographical coordinates.
In part 2 ‘status of schistosomiasis in the Caribbean’, it would be much improved if current environmental and social conditions are provided, such as average rainfall, average height, population, and human lifestyle.
In ‘2.2. Localities where eliminated but not yet verified’ and ‘2.3. Localities where still considered ongoing’, time tables with each control method or program and their contents should be clarified to show the changes in prevalence and effects.
Line 225-227 of ‘2.2.2. Dominican Republic’, the conclusion part starting in line 225 (Evidence suggests…of competing snails) needs supporting reference(s).
In line 58, the sentence ‘There a few islands where schistosomiasis was endemic that…’ is not grammatical.
Round 2
Reviewer 3 Report
This manuscript has been improved, except for missing a map.